# How We Treat Localized Rectal Cancer—An Institutional Paradigm for Total Neoadjuvant Therapy

**DOI:** 10.3390/cancers14225709

**Published:** 2022-11-21

**Authors:** Falk Roeder, Sabine Gerum, Stefan Hecht, Florian Huemer, Tarkan Jäger, Reinhard Kaufmann, Eckhard Klieser, Oliver Owen Koch, Daniel Neureiter, Klaus Emmanuel, Felix Sedlmayer, Richard Greil, Lukas Weiss

**Affiliations:** 1Department of Radiation Oncology, Paracelsus Medical University Salzburg, 5020 Salzburg, Austria; 2Department of Radiology, Paracelsus Medical University Salzburg, 5020 Salzburg, Austria; 3Department of Internal Medicine III with Haematology, Medical Oncology, Haemostaseology, Infectiology and Rheumatology, Oncologic Center, Salzburg Cancer Research Institute—Laboratory for Immunological and Molecular Cancer Research (SCRI-LIMCR), Center for Clinical Cancer and Immunology Trials (CCCIT), Cancer Cluster Salzburg, Paracelsus Medical University Salzburg, 5020 Salzburg, Austria; 4Department of Visceral and Thoracic Surgery, Paracelsus Medical University Salzburg, 5020 Salzburg, Austria; 5Institute of Pathology, Paracelsus Medical University Salzburg, Cancer Cluster Salzburg, 5020 Salzburg, Austria

**Keywords:** total neoadjuvant therapy, TNT, rectal cancer, radiotherapy, chemoradiotherapy, CRT, organ-sparing, mismatch-repair-deficiency, microsatellite instability

## Abstract

**Simple Summary:**

Treatment of locally advanced rectal cancer has been subject to pronounced changes based on the results of recent prospective trials. New paradigms have been introduced, including the shift of systemic treatment components from adjuvant to neoadjuvant setting (total neoadjuvant therapy), the omission of surgery in patients with clinical complete responses after neoadjuvant treatment (non-operative management) and the introduction of upfront immunotherapy in patients with microsatellite instability (MSI)-high/mismatch-repair-deficient (dMMR) tumors. We developed an institutional treatment algorithm which may serve as a practical tool for treating physicians without any claim to general validity.

**Abstract:**

Total neoadjuvant therapy (TNT)—the neoadjuvant employment of radiotherapy (RT) or chemoradiation (CRT) as well as chemotherapy (CHT) before surgery—may lead to increased pathological complete response (pCR) rates as well as a reduction in the risk of distant metastases in locally advanced rectal cancer. Furthermore, increased response rates may allow organ-sparing strategies in a growing number of patients with low rectal cancer and upfront immunotherapy has shown very promising early results in patients with microsatellite instability (MSI)-high/mismatch-repair-deficient (dMMR) tumors. Despite the lack of a generally accepted treatment standard, we strongly believe that existing data is sufficient to adopt the concept of TNT and immunotherapy in clinical practice. The treatment algorithm presented in the following is based on our interpretation of the current data and should serve as a practical guide for treating physicians—without any claim to general validity.

## 1. Introduction

For two decades, locally advanced rectal adenocarcinoma was approached by neoadjuvant short- or long-course radiation therapy (RT) with or without concomitant CHT followed by surgery and—in case of pathologic nodal involvement—adjuvant CHT. This approach was based on proven efficacy and acceptable toxicity in large, randomized trials [1,2,3]. The combination of the neoadjuvant approach with modern surgical techniques such as total mesorectal excision (TME) led to a distinct decrease in locoregional failures. Other limitations moved into the focus of further clinical development: The need for permanent colostomies in distal tumors or the respectively poor functional outcome of very low anastomoses prompted investigators to evaluate a selective organ-preserving non-operative management (NOM) in patients with good response to neoadjuvant CRT. This strategy of so called “watch and wait” with close follow-up after clinical complete remission (cCR) comprising the option of salvage surgery in case of a locoregional recurrence was pioneered by Habr-Gama et al. [4] and continued by others [5,6,7]. They consistently showed high rates of organ-preservation (26–58%) without compromising overall survival (OS) based on non-randomized data, highlighting the general possibility of this approach in reminiscence of anal cancer [8]. In parallel, several study groups evaluated the issue of high distant failure rates by transferring systemic treatment into the neoadjuvant phase, known as total neoadjuvant therapy (TNT). There is currently no generally agreed definition of the term TNT. For the scope of this article, we define TNT as a neoadjuvant regimen that includes a radiation therapy part (either RT alone or chemoradiation) and a chemotherapy part, both applied prior to surgery regardless of their sequence.

Recently, two large randomized phase III trials have reported significantly improved distant failure and disease-free survival (DFS) rates using either induction or consolidation CHT sequential to (chemo)radiation compared to the standard approach [9,10]. Aside from lowering the risk of distant failure, both trials consistently showed nearly doubled pCR rates (12–14% vs. 28%), further enhancing the possibilities for organ-sparing concepts [9,10]. Both strategies were combined in the OPRA trial [11], which randomized patients to a moderately dose-intensified neoadjuvant CRT regimen (54 Gy) either preceded or followed by CHT while selectively withholding surgery in case of complete clinical remission. As already assumed, based on the results of its predecessor study [12] and a German randomized phase II trial [13], consolidation CHT resulted in higher organ preservation rates compared to induction CHT.

However, aside from similar major findings, the mentioned trials still considerably differed regarding the used (chemo)radiation regimen and several inclusion criteria. Especially the risk profile for locoregional and/or distant failure based on certain prognostic factors at presentation, such as mesorectal fascia involvement (MRF+), extent of nodal involvement (lateral vs. perirectal), or the presence of extramural vascular invasion (EMVI), was not uniform across the trials. This results in some uncertainty regarding the optimal treatment approach for different risk groups of patients: who should be scheduled for a TNT approach per se, which (chemo)radiation regimen should be included for whom, who should be selected for organ preservation, and how should the follow-up be performed in these patients. At our institution, changes in the general treatment algorithm were preceded by intensive, structured multidisciplinary discussions of the available data including specialized surgeons, radiation and medical oncologists, radiologists, and pathologists, resulting in a comprehensive consensual new treatment algorithm. The current manuscript aims at presenting this algorithm including its rationale based on our interpretation of the available literature.

## 2. Decision Making

Within a TNT and/or NOM concept, initial staging gains even more importance for patient selection due to the transfer of all additional treatments into the preoperative phase and the increasing number of prognostic factors with relevance for treatment decisions. The most relevant factors for defining risk group and treatment strategy include T and N stage, depth of infiltration into the mesorectal tissue, involvement of the mesorectal fascia, presence of lateral (extramesorectal) suspicious lymph nodes, presence of EMVI, and tumor distance from the anal verge. All the mentioned factors are known to be associated with the risk of either locoregional recurrence or distant metastases or (mainly) both [14,15,16,17,18,19,20,21]. Distal tumor location may additionally result in the necessity of performing an abdominoperineal resection with permanent colostomy. Aside from its impact on quality of life, abdominoperineal resection itself has been shown to be associated with an increased risk of local and distant failure, as well as shorter OS when compared to low anterior resection [14].

Based on current recommendations, these factors are determined most accurately by a combination of pelvic MRI and rigid endoscopy with endorectal ultrasound (EUS), flanked by CT of chest and abdomen or FDG-PET-CT to rule out distant metastases [22]. While EUS is superior to MRI in differentiating T1 and T2 tumors [23], MRI holds a clear advantage in accurate definition of higher T stages (especially regarding involvement of the MRF and adjacent organs) and defining the relationship to the sphincter complex [24,25], although it may overstage some T2 tumors due to desmoplastic reaction [25]. MRI can further most precisely assess the depth of invasion into the mesorectal fat and therefore allows subclassification of T3 stages (at least into <5 mm vs. ≥5 mm, which is most important for risk assessment) [24]. The key strength of MRI staging is that it allows the precise visualization of the mesorectal fascia in pretreatment situation and after neoadjuvant therapy [23,26], which is of major interest regarding assessment of circumferential resection margin (CRM) involvement for patients undergoing TME surgery [27]. The presence of EMVI can also be predicted most accurately by MRI [22] and its detection on MRI has been shown to be adversely correlated with distant control [25]. In contrast, nodal staging remains challenging and lacks sensitivity in all imaging modalities [24]. Although its value as a prognostic factor for local recurrence (LR) is less pronounced since the introduction of TME (if any) [24], (lateral) nodal positivity is still associated with increased risk for locoregional and distant failure [15,18]. Given the increased sensitivity and specificity with the adoption of more sophisticated criteria for nodal staging (as outlined for example in the ESGAR recommendations) [22], we decided to include nodal stage based on MRI into our decision making.

Regarding response evaluation, standard MRI alone lacks accuracy for precisely defining ycT and ycN stage [23] and prediction of complete remissions, which in some reports is even lower than with endoscopy and digital rectal examination (DRE) [23]. However, the accuracy of MRI can be improved by the addition of diffusion-weighted-imaging (DWI) sequences, which allow for a better differentiation between residual tumor and fibrosis [23], and a combination of all modalities can even achieve superior results [17,22]. We therefore decided to use a combination of endoscopy, DRE, and MRI (including DWI) for response evaluation, especially for decisions regarding a watch and wait approach.

Technically, there is broad consensus that T2 weighted sequences acquired in three planes including thin axial sections (slice thickness 3 mm) perpendicular to the long axis of the rectal wall at tumor location build up the basis for accurate rectal cancer staging [25,28]. Although the use of contrast is debatable, some work suggests increased accuracy in detecting adjacent organ infiltration and possible value in the evaluation of internal changes in tumor morphology during response assessment [25]. DWI sequences may improve the prediction of involved lymph nodes at presentation [25] and increasing evidence exists for its value in distinguishing between partial and complete response during response assessment [22,29]. The MRI protocol used at our institution mainly follows the ESGAR recommendations [22] and includes high resolution T2-weighted sequences through the tumor as mentioned, large field of view axial T2 weighted sequences of the whole pelvis, DWI sequences, and contrast enhanced T1-weighted sequences. The same MRI protocol is used for primary staging and response evaluation to ensure comparability. Scans are performed on either 1.5 Tesla Philips Ingenia or 3 Tesla Philips Achieva Scanners (Philips Healthcare^®^, Best, The Netherlands). We routinely use spasmolytics such as butylscopolamin or glucagon to reduce artefacts from peristalsis, but discourage the routine use of rectal fillings due to possible misinterpretation of the distance between the tumor and the mesorectal fascia with the risk of overstaging [25,30].

Regarding the detection of distant metastases, current guidelines recommend the use of chest and abdominal CT [22,24]. FDG-PET-CT might be superior to CT staging in the detection of metastatic disease beyond the liver [26], with some authors reporting changes in patient management after FDG-PET-CT in 8–11% of patients [26]. We therefore decided to integrate FDG-PET-CT into the routine staging at least in patients scheduled for TNT approaches, because those patients will be at the highest risk for undetected distant spread. 

In summary, we strongly acknowledge the importance of extensive primary staging and sophisticated response evaluation in the decision-making process. We therefore decided to perform all staging procedures (including rigid endoscopy with EUS, pelvic MRI, and FDG-PET-CT) and the final response assessment after completion of neoadjuvant treatment preferably at our tertiary cancer center. If a NOM approach is used, all follow-up examinations for the first two years are also scheduled at our institution. We further agreed on criteria for the standardized assessment and reporting of pelvic MRI during staging, response assessment, and follow-up together with our radiologists. These criteria are mainly based on the ESGAR recommendations [22] for MRI and response criteria used in the OPRA trial [31] (Table 1 and Table 2). Patients will only be scheduled for a TNT approach if deemed fit for doublet chemotherapy and if TNT is unlikely to cause excessive toxicity, which could endanger ensuing surgery in curative intent. Otherwise, patients will be evaluated separately by all participating disciplines (medical oncology, radiation oncology, and surgery) and counseled individually after a second multidisciplinary discussion. 

## 3. Treatment Algorithm

### 3.1. General Remarks

Our TNT-approach is mainly based on the regimens used in the RAPIDO- (short-course radiation therapy (RT) followed by consolidation CHT) [10] and OPRA-trial (long-course CRT) followed by consolidation CHT) [11] prior to surgery or watch and wait. Although we do acknowledge the positive results obtained in the PRODIGE 23 trial (induction FOLFOXIRI followed by long-course CRT prior to surgery and adjuvant CHT) [9]—we deem this CHT regimen too toxic for most cases of rectal cancer, especially considering the risk of clinical overstaging and, consequently, of overtreatment. In addition, there is no clear evidence for the benefit of irinotecan in the (neo-)adjuvant setting in colorectal cancer [32] or for adding adjuvant CHT after surgery following a TNT approach. Cross-trial comparison of the PRODIGE 23 and RAPIDO trials does not suggest superiority of FOLFOXIRI followed by long-course chemoradiotherapy [9] over short-course radiotherapy followed by CHT with CAPOX or FOLFOX [10] with regard to pCR rate or distant failure. Moreover, both randomized trials comparing different sequences of TNT approaches (RT followed by consolidation CHT vs. induction CHT followed by RT) favored the RT-first approach in terms of either increased pCR rates, TME-free survival, or reduced side effects [11,13]. Comparing the RT parts within both TNT concepts, we generally assume a slightly increased likelihood of tumor cell death and, consequently, a downsizing with long-course CRT compared to short-course RT. This hypothesis is based on radiobiological assumptions (higher biologically effective dose (BED) of long-course RT according to the linear-quadratic model) [33] and clinical data from non-TNT approaches [34]. 

In our opinion, multimodal treatment should follow two major rules: risk-adaption and preservation of quality of life, leaving us with three major patient groups: Patients with high-risk factors;Patients with distal tumors;Patients with intermediate-risk factors and non-distal tumors.

### 3.2. Presence of High-Risk Factors

We defined high-risk features on pelvic MRI according to the inclusion criteria of the RAPIDO trial [10]: cT4 stage, involvement of the mesorectal fascia (MRF+), extramural vascular invasion (EMVI+), and presence of suspicious lateral (extramesorectal) regional lymph nodes. Although not unique as prognostic parameters in rectal cancer, those factors clearly represent situations with distinctly increased risk for locoregional and/or distant failure. The strong association of T stage with the outcome has been described for decades and has been confirmed in large databases irrespective of treatment. For example, Bernstein et al. [15] analyzed roughly 2800 patients treated by surgery only from a Norwegian registry, while Valentini et al. [14] evaluated a similar number treated with neoadjuvant (chemo)radiation in a pooled analysis of five European trials, both showing significantly decreased LC and OS with increasing T stage. Involvement of the mesorectal fascia (MRF+), which can be accurately assessed by MRI [23] and correlates with positive circumferential margins after resection, is also a strong negative prognostic factor for LC and OS according to a recent meta-analysis including 17,500 patients [35]. The presence of EMVI on MRI has been shown to result in reduced distant control and DFS based on meta-analyses [19] and prospective data [36]. Finally, lateral lymph nodes (defined as extramesorectal, regional lymph nodes typically located in the iliac region) have been identified as a risk factor for locoregional recurrence [37] and likely represent an increased risk for distant failure.

Because these patients are at the highest risk for locoregional and/or distant failure, we offer them the most intensive therapy within our TNT concept, namely long-course CRT followed by consolidation CHT (Figure 1) and surgery or NOM based on tumor location and response.

### 3.3. Distal Tumors

Patients with distal tumors are at the highest risk for the need of an abdominoperineal resection with permanent colostomy, limiting their quality of life. Moreover, they face a considerable risk of functional impairment even after sphincter-saving procedures known as low anterior resection syndrome (LARS). A recent meta-analysis evaluated 36 trials and quantified the risk for LARS after low anterior resections as high as 44% [38]. Thus, patients with distal tumors may benefit the most from organ-preserving approaches, as shown by early data comparing the quality of life between patients assigned to NOM after cCR or surgery with pCR, favoring NOM in nearly all functional parameters [6]. Aside from quality-of-life aspects, abdominoperineal resections have been shown to be an independent risk factor for a worse outcome in terms of LC and OS [16]. However, based on the currently available data, NOM approaches only seem safe in patients achieving a cCR after neoadjuvant treatment [39]. Thus, a strategy aiming at the benefits from organ preservation should use a neoadjuvant treatment with a high likelihood of achieving a cCR.

We therefore offer all patients with distal tumors the most intensive therapy within our TNT concept, namely long-course CRT followed by consolidation CHT and NOM if a cCR is achieved (Figure 2). In patients with incomplete response of the primary tumor following TNT, transanal local excision should be evaluated if no other suspicious lesions are found on restaging (especially no lymph nodes). In the absence of a generally agreed, easily assessable surgical definition of sphincter-sparing resectability, we defined all tumors with a lower edge within 5 cm from the anal verge (based on rigid endoscopy) or deemed not eligible for sphincter-preserving surgery at baseline (based on evaluation by an experienced surgeon) as distal. This strategy is offered even to patients suffering from early-stage rectal cancer (who otherwise would not have received neoadjuvant treatment at all) after individual counseling, if sphincter-sparing surgery seems impossible. 

### 3.4. Non Distal, Non High-Risk Tumors

Excluding patients with high-risk features and distal tumors leads to a third group of patients suffering from rectal cancer with “intermediate”-risk features located in the middle and upper third of the rectum. Several non high-risk but still prognostic factors exist to guide treatment decisions in those patients. 

Tumor location in the upper third as a factor has been a matter of debate since the introduction of neoadjuvant treatment, based on the different definitions of rectal length in US and European trials [40]. Data on the question if upper third rectal cancers behave more like colon or lower rectal cancers and should be treated according to one or the other are still conflicting. However, based on recent trials and large population-based analyses, the relative effect of neoadjuvant RT in lowering the LR risk seems widely independent from tumor height (at least if confounding factors such as CRM positivity are excluded) [2,21,41]. However, the absolute risk for LR seems considerably lower in upper third tumors compared to more distal locations [20]. Therefore, the relative benefit from the addition of RT becomes nearly meaningless from a clinical point of view, especially in the absence of high-risk factors. Therefore, we offer those patients (with a tumor located 10–15 cm from the anal verge, no high-risk features, cN0-1) an upfront surgery approach followed by adjuvant CHT based on pathological lymph node stage similarly to colon cancer (Figure 2). The only exemption are patients with a high number of involved lymph nodes (N2 situation) based on pretreatment MRI. Even assuming the difficulties in correctly predicting nodal stage on MRI, those patients will likely experience a need for additional CHT to reduce their increased risk of distant failure. Moreover, based on our clinical experience, those patients often suffer from involved lymph nodes distal from their primary tumor, likely increasing the risk for LR. Indeed, some evidence points at an increased risk for LR in patients with high nodal load even in the era of TME surgery [42]. We therefore offer those patients (tumor located 10–15 cm from the anal verge, cN2 situation) the RAPIDO concept with short-course RT followed by consolidation CHT and surgery (Figure 2). 

Another factor predicting the outcome is the depth of mesorectal invasion. This knowledge has led to a subclassification of T3 stage (which is still not part of the UICC and AJCC staging manuals) into four stages: T3a (<1 mm invasion), T3b (≥1–5 mm), T3c (>5–15 mm), and T3d (>15 mm depth of invasion) [43]. While the impact of mesorectal invasion on outcome (especially LR risk) has been consistently shown in different studies [44,45], data on the best discrimination level between lower and higher risk is still conflicting. Most studies favor a two-stage approach with an invasion depth around 5 mm (4–6 mm) to be discriminative enough for clinical purposes [44,45]. We therefore offer patients with tumors located in the middle third and cT3c/d stage a neoadjuvant approach consisting at least of preoperative short-course RT because of their increased risk for LR (Figure 2), while the addition of consolidation CHT is based on nodal status (see below). 

Lymph node status represents one of the oldest but most debated risk factors in rectal cancer. Its general impact on outcome seems crystal-clear based on large, pooled analyses [14,46] and prospective trials [2,21]. For example, Shen et al. [46] analyzed >8000 patients treated by surgery only and observed a significant stepwise decrease in 5-year cancer-specific survival from 80% in pN0 patients to 53% in pN2b patients. The MRC CR07 trial [21] showed a clear association between pathological nodal status and LR risk in both treatment arms, while the Dutch rectal cancer trial [2] showed a similar association between pathological nodal status and OS (although according to unplanned post-hoc subgroup analyses in both trials). However, the value of nodal status for treatment decision making is highly debated because of the difficulties in adequately assessing nodal status prior to surgery based on preoperative imaging alone, which may lead to overtreatment. Nevertheless, Valentini et al. [14] found a significant association between pretreatment nodal status (based on MRI) and DFS as well as OS in a pooled analysis of five European trials using neoadjuvant (C)RT. Patients with positive nodes had a significant benefit regarding OS in the Dutch rectal cancer trial [2] and regarding LC in the MRC CR07 trial [21] after neoadjuvant treatment compared to surgery alone or selective postoperative CRT (again according to unplanned post-hoc subgroup analyses). 

While overtreatment may represent a considerable risk, undertreatment caused by inaccurately described nodal negativity might be even worse. Several analyses have shown rates of pathologically confirmed positive nodes in 20–30% of patients initially staged as cT3cN0 on EUS and MRI with or without neoadjuvant treatment [47,48,49]. In patients treated by surgery alone, understaging seemed more frequent than overstaging [47], thus making undertreatment more likely than overtreatment. While there is some evidence that MRI nodal staging accuracy can be improved by using more sophisticated MRI criteria [22], it remains an issue of treatment philosophy if a possibility of under- or overtreatment seems more harmful to the patient, which needs to be individually discussed. In summary, we decided to still incorporate MRI predicted nodal status as a factor in treatment decision making (despite its suboptimal accuracy), but in addition to other factors like depth of invasion and tumor location and with regard to the potential risk of both local and distant failure. We therefore offer patients with tumors located in the middle third upfront surgery only in cases of less than 5 mm depth of mesorectal invasion (cT3a/b on MRI) and in the absence of MRI predicted nodal spread (cN0). Patients with >5 mm depth of mesorectal invasion are offered a neoadjuvant concept with at least short-course RT if node-negative on MRI, followed by consolidation CHT (RAPIDO-like) if node-positive on MRI (Figure 2).

## 4. Response Assessment

Similarly to the increasing importance of pre-treatment evaluation for the indication of neoadjuvant therapies since the introduction of TNT concepts, response evaluation has gained critical importance in the view of NOM. As NOM-approaches based on current data only seem to be safe after achieving a cCR [39], all possibilities to ensure an adequate detection of cCR should be exhausted. As mentioned above, we therefore decided to use a combination of DRE, standardized MRI, endoscopy and FDG-PET-CT all performed in-house to ensure proper patient selection for NOM. While ruling out new distant metastases seems reasonably achieved by FDG-PET-CT, we identified three major issues of concern regarding the detection of a “locoregional” cCR during our initial discussions: Which specific parameters to define a local cCR on imaging and endoscopy should be used?Which is the best timing for response assessment?Do we attempt a histological “confirmation” (e.g., biopsy) of a cCR?

The general advantages and disadvantages of different MRI sequences in response assessment have already been discussed above. In the absence of clear evidence (e.g., based on direct prospective comparisons) favoring certain parameters for classification of a response as “complete” or the prediction of a pCR based on MRI and/or endoscopy, we decided to simply follow the criteria for cCR of the Memorial Sloan Kettering Cancer Center (MSKCC) regression scheme used in the OPRA trial (the largest prospective trial evaluating a NOM-approach) [11,31] as the best available evidence (Table 1). In contrast to the OPRA scheme, we favor a two-tiered approach only differentiating between cCR and non-cCR, which refers to the primary tumor and all initially suspicious pelvic nodes. Only patients with a cCR of all suspicious lesions (ycT0 ycN0) will be offered a NOM-approach. 

The optimal time interval for response assessment is highly debated. For non-TNT approaches, there is increasing evidence that longer time intervals after (chemo)radiation (>8 weeks) may increase the likelihood of pCR without facing higher postsurgical morbidity [50,51]. In contrast, a randomized trial specifically addressing this issue by comparing surgery after 7 weeks with 11 weeks from the end of neoadjuvant CRT failed to show any benefit from a longer waiting time with regard to pCR rates, but reported detrimental effects on postoperative morbidity [52]. Major trials using TNT concepts also differed distinctly with regard to the time interval between the end of neoadjuvant therapy and surgery. While the RAPIDO investigators [10] chose a short interval of 2–4 weeks to surgery in their TNT arm, the PRODIGE 23 [9] and CAO/ARO/AIO-12 [13] investigators favored a conservative interval of 6–8 weeks and the OPRA investigators [11] even waited 8 weeks for response evaluation in both arms to decide whether a patient needs surgery or not. With regard to the time interval between the end of radiotherapy to surgery, the differences are even larger due to the different scheduling (PRODIGE 23: 6–8 weeks [9], CAO/ARO/AIO-12: 6.5 or 13 weeks [13], RAPIDO: 23–25 weeks [10], OPRA: 8 or 26–28 weeks [11]). In the absence of clear evidence favoring one interval over the others, we decided to schedule response evaluation roughly 4 weeks after completion of consolidation CHT for practical reasons, followed by surgery (if at all) within 2–4 weeks. This would equal a time interval of 6–8 weeks from the end of neoadjuvant treatment to surgery similarly to PRODIGE 23 and CAO/ARO/AIO-12 and stays in between the range of the other trials. Regarding the time interval between the end of RT and surgery, this would equal 24–28 weeks similarly to the RAPIDO and OPRA trials, which represent the basis of our TNT approach. We might underestimate the putative rate of NOM candidates, due to the slightly shorter interval to response evaluation compared to OPRA. However, we feel that this difference is most likely of minor importance as downsizing induced by radiation likely needs more time than by CHT, and the interval from the end of RT already seems extended. For patients receiving sole neoadjuvant short-course RT, we adhere to our previous standard of performing surgery within 1 week as no downstaging is anticipated, and a recent analysis comparing immediate versus delayed surgery after short-course RT found increased LR rates with delayed surgery [53]. 

To ensure maximum safety for patients within NOM approaches, some centers advocated (blind or guided) biopsies of the primary tumor region to “confirm” a complete remission in contrast to relying only on endoscopy and imaging. After multidisciplinary discussion, we decided not to perform such biopsies for the following reasons: We offer a NOM approach only in case of a cCR of primary tumor and suspicious nodes (ycT0ycN0). As the latter are mostly not amendable to biopsy, we will have to rely on imaging in most patients anyway. Moreover, the combination with endoscopy already increases safety with regard to possible superficial residuals of the primary tumor, and blind biopsies of the tumor region in case of complete remissions seem generally flawed by the risk of geographic misses [29]. Because of the excellent existing salvage surgery opportunities especially for superficial endoluminal recurrences and the close follow-up with repeated MRI and endoscopy (which includes biopsy of suspicious progressive lesions), we have decided to waive such procedures and rely solely on DRE, endoscopy, and MRI imaging during response evaluation. 

## 5. Follow-Up

Our follow-up procedure generally depends on whether planned surgery has been done or if the patient is part of a NOM approach. Surgically treated patients are followed according to institutional standards based on current guidelines, which will not be elaborated any further. For patients within a watch and wait policy after cCR, we opted for a very close follow-up scheme (Table 2). This scheme is generally based on the MSKCC approach [31] used in the OPRA trial [11] including mainly a combination of DRE, endoscopy, and MRI of the pelvis for local assessment and CT of the thorax and abdomen for distant assessment. As most recurrences occur in the first two years [11], we opted to even increase the intensity of follow-up examinations especially in this time frame. 

## 6. Mismatch-Repair-Deficiency (dMMR)/Microsatellite Instability (MSI-High)

Around 5 to 10% of rectal cancers show microsatellite instability (MSI-high) or mismatch-repair–deficiency (dMMR) [54]. This subtype of colorectal cancer has shown exquisite sensitivity to immune checkpoint blockade with superior response, PFS and OS in the metastatic setting when compared to standard systemic therapy [55]. Therefore, treatment with the anti-PD-1 antibody pembrolizumab has become the standard first line of treatment for MSI-high/dMMR metastatic colorectal cancer.

Recently, first data have been presented for treating locally advanced rectal cancer patients with the anti-PD-1 antibody dostarlimab [56]: in a single-arm phase II study patients with clinical stage II and III dMMR rectal cancer were treated for 6 months with dostarlimab. In the first 14 patients evaluable, cCR—defined by MR, endoscopy and digital rectal examination—could be achieved in 100% of patients. Patients with a cCR were followed in a watch and wait strategy and were neither subjected to (chemo)radiation nor surgery. These results are also corroborated by other studies, with evidence of high clinical efficacy of immune checkpoint blockade in the treatment of localized colorectal cancer [57,58].

Although these results have to be confirmed in larger trials before entering clinical routine, we believe that in the light of these data, treatment with immune checkpoint blockade should be discussed with patients with MSI-high/dMMR rectal cancer on an individual basis, especially when aiming for an organ-sparing strategy. We strongly encourage referring these patients to centers with respective clinical trials, in order to contribute to the body of knowledge that will eventually lead to the approval of these drugs and thereby the possibility of universal employment in all patients with MSI-high/dMMR disease.

## 7. Conclusions

In light of increased pCR rates and improved DFS, we have implemented total neoadjuvant therapy (TNT) as institutional standard for treating patients with locally advanced rectal cancer in the presence of high-risk features. Furthermore, we routinely offer TNT to patients with distal rectal cancer with the goal of non-operative management. Whenever TNT is indicated, immunotherapy should be considered in case of microsatellite instability. The presented treatment algorithm is based on our interpretation of the current data and should serve as a practical guide for treating physicians—without any claim to general validity.

## Figures and Tables

**Figure 1 cancers-14-05709-f001:**
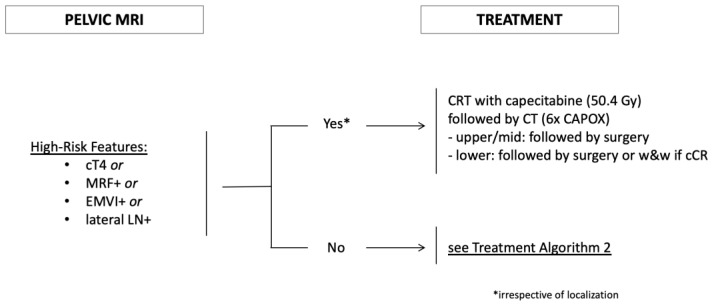
Treatment Algorithm 1: Risk-stratified treatment of rectal cancer. MRF+ (involvement of mesorectal fascia); EMVI+ (presence of extramural vascular invasion); lateral LN+ (suspicious lateral pelvic lymph nodes, located along the iliac and obturator arteries); CRT (chemoradiation); CT (chemotherapy).

**Figure 2 cancers-14-05709-f002:**
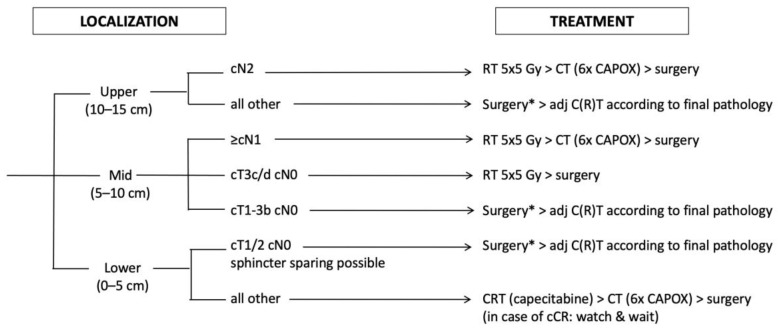
Treatment Algorithm 2: treatment in the absence of high-risk features. Localization is given as the distance from the anal verge; RT (radiation therapy); CT (chemotherapy); C(R)T (chemo(radio)therapy); cCR (clinical complete remission); * cT1cN0 tumors may be treated with local excision in selected cases.

**Table 1 cancers-14-05709-t001:** Definition of complete clinical remission following TNT (Memorial Sloan Kettering Regression Scheme).

Digital Rectal Examination	Normal
Endoscopy	Flat, white scar
	Telangiectasia
	No ulcer
	No nodularity
MR-T2W	Only dark T2 signal
	No intermediate T2 signal
	AND
	No visible lymph nodes
MR-DW	No visible tumor on B800 to B1000 signal
	AND/OR
	Lack of OR low signal on ADC map
	Uniform, linear signal in wall above tumor is acceptable

MRI (magnetic resonance imaging); T2W (T2-weighted image); DW (diffusion-weighted image); ADC (apparent diffusion coefficient).

**Table 2 cancers-14-05709-t002:** Follow-up scheme for non-operative management (NOM).

	Year 1				Year 2				Year 3				Year 4				Year 5			
Month	3	6	9	12	3	6	9	12	3	6	9	12	3	6	9	12	3	6	9	12
Clinical Evaluation	X	X	X	X	X	X	X	X		X		X		X		X		X		X
DRE	X	X	X	X	X	X	X	X		X		X		X		X		X		X
CEA	X	X	X	X	X	X	X	X		X		X		X		X		X		X
Proctoscopy	X	X	X	X	X	X	X	X		X		X		X		X		X		X
MR pelvis	X	X	X	X	X	X	X	X		X		X		X		X		X		X
CT thorax & abdomen		X		X		X		X		X		X				X				X
Colonoscopy				X												X				

DRE (digital rectal exam); CEA (carcinoembryonic antigen); MR (magnetic resonance imaging); CT (computed tomography).

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
