# Peer review of "How We Treat Localized Rectal Cancer—An Institutional Paradigm for Total Neoadjuvant Therapy"

_cancers, 2022, doi:10.3390/cancers14225709_

Round 1
Reviewer 1 Report
Thank you very much for this interesting and useful article. I think in every tertiary hospital there is some discussion during MDT meeting about these patients. This article is practical point of view of treating patients with localized rectal cancer. It is not really scientific or review article in my opinion, so it is not ethical for me to comment or make suggestions to other doctors on their diagnostic or treatment algorithms or concepts, they can be different from hospital to hospital, or country to country, especially when we talk about rectal cancer. Treatment of rectal cancer is very individual in my opinion. So my comments or questions are more practical. 1. the description and definition of total neoadjuvant treatment should be more detailed in my opinion. 2. High risk features: MRF+ and T4 are similar ( or same) criteria in my opinion, if its cT4, it's always MRF+, or isn't? 3. Is there really benefit for such aggressive neoadjuvant treatment (TNT) for small tumors (T2, T3) and suspicious (because it's always just suspicion radiologically..) EMVI + and/or lateral lymph nodes +. And what is the aim in this case? Sterilization of these nodes? how can we control that? What are surgeons doing during surgery with these nodes? 4. what about the age of the patient and the comorbidities? Are they changing decision making? 5. Is there place for ChRT without adding CHT in neoadjuvant treatment protocols( if you have good response, MRF- after ChRT). 6. Is there place for neoadjuvant CHT without RT in neoadjuvant treatment? Most clear indication for neoadjuvant treatment for rectal cancer is necessity of tumor downstaging and downsizing ( MRF+ or sphincter sparing for low cancers) and this has to be discussed individually with patient and in MDT meeting. So in my personal opinion there is possibility of overtreatment with this TNT approach for many groups of patients, according this concept. But that's just my personal opinion as mentioned above, the article is really useful and valuable.
Author Response
“Thank you very much for this interesting and useful article. I think in every tertiary hospital there is some discussion during MDT meeting about these patients. This article is practical point of view of treating patients with localized rectal cancer. It is not really scientific or review article in my opinion, so it is not ethical for me to comment or make suggestions to other doctors on their diagnostic or treatment algorithms or concepts, they can be different from hospital to hospital, or country to country, especially when we talk about rectal cancer. Treatment of rectal cancer is very individual in my opinion.”
Answer: We thank the reviewer for his (or her) open-minded answer and for respecting our thoughts regarding our proposed treatment concept as an individual interpretation of the available data.
“So my comments or questions are more practical. 1. the description and definition of total neoadjuvant treatment should be more detailed in my opinion.“
Answer: We are not quite sure what the reviewer is addressing: Currently, there is no generally agreed definition of the term “total neoadjuvant treatment”. In rectal cancer the term is mainly used to describe treatment approaches which apply all additional treatments (e.g. radiation therapy, systemic therapy) prior to surgery in contrast to the previous standard approaches, which usually include adjuvant (postoperative) chemotherapy. In a less general sense, the term is restricted to neoadjuvant treatment regimens which include a radiotherapy-containing part (either RT alone or chemoradiation) and a sequential chemotherapy part. Both parts are applied prior to surgery (while it does not matter which comes first) and no adjuvant therapy is given. For the scope of this article we refer to the term TNT in the latter sense. For clarification, we have included the following description into the introduction.
“There is currently no generally agreed definition of the term TNT. For the scope of this article, we define TNT as a neoadjuvant regimen which includes a radiation therapy part (either RT alone or chemoradiation) and a chemotherapy part, both applied prior to surgery regardless of their sequence.“
“2. High risk features: MRF+ and T4 are similar ( or same) criteria in my opinion, if its cT4, it's always MRF+, or isn't?”
Answer: We agree with the reviewer that cT4 tumors are usually MRF+, however this needs not to be true vice versa. According to the TNM staging manual, cT4 is restricted to tumors either invading the visceral peritoneum (which is usually found mainly in proximal rectal cancers) or invasion/adherence into adjacent organs or structures (like prostate, seminal vesicles, cervix or vagina), which might be found in mid or distal rectal cancers. Interestingly, even the extended staging manual does not provide clear suggestions if invasion of the sphincter muscles has to be staged as cT4. Therefore, many tumors with MRF+ (which is defined as invasion or by less than 1 mm distance between tumor and fascia) do not fulfill the formal definition of cT4 tumors, although their biological behavior might be similar. Moreover, the Dutch RAPIDO trial used T4 and MRF+ as separate inclusion criteria. We therefore agree, that many tumors may show both features, however we feel that both risk factors should be taken into account separately as they are not fully congruent.
“3. Is there really benefit for such aggressive neoadjuvant treatment (TNT) for small tumors (T2, T3) and suspicious (because it's always just suspicion radiologically..) EMVI + and/or lateral lymph nodes +. And what is the aim in this case? Sterilization of these nodes? how can we control that? What are surgeons doing during surgery with these nodes?”
Answer: We agree with the reviewer that assessment of risk factors prior to surgery by MRI is not completely accurate and may result in some over- (or under-) treatment. However, all of the mentioned factors have been shown to be independent prognostic factors for locoregional and/or distant control in meta-analyses, randomized trials or large cohort studies. Moreover, all mentioned risk factors (including EMVI and presence of lateral nodes) have been used as separate inclusion criteria within the RAPIDO trial, which showed a clear DFS benefit for a TNT approach compared to the standard approach in the entire study cohort. It cannot be concluded from the available data, if TNT resulted in an increased sterilization rate of lateral nodes if no other risk factors are present because no corresponding subgroup analyses have been published so far. Nevertheless, based on the results of the RAPIDO trial, there is a potential metastasis-free- and disease-free-survival benefit for these patients even in the absence of other risk factors, which suggests the possibility of increased sterilization rates of such deposits. We therefore feel that treatment with our concept is justified based on the RAPIDO data.
Regarding the surgical consequences: Control is similar to all other factors during response evaluation. For example, cCR triggering a watch and wait approach in distal rectal cancers includes the remission of a prior visible EMVI and lateral nodes. According to local surgical standard, patients with pathologically enlarged lateral pelvic lymph nodes on initial MR staging will undergo respective extended dissection of lateral pelvic lymph nodes in case of non cCR. In case of cCR non operative management may be offered. Given the scope of our manuscript we did not elaborate on specific surgical methods.
“4. what about the age of the patient and the comorbidities? Are they changing decision making?”
Answer: We thank the reviewer for this comment. We do not use age per se for decision making, but of course performance status, comorbidities and/or frailty impact the decision-making process. Patients, who are deemed unfit for doublet chemotherapy will not be scheduled for a TNT approach. We have included the following paragraph into the decision-making part.
„Patients will only be scheduled for a TNT approach if deemed fit for doublet chemotherapy and if TNT is unlikely to cause excessive toxicity, which could endanger ensuing surgery in curative intent. Otherwise, patients will be evaluated separately by all participating disciplines (medical oncology, radiation oncology and surgery) and counseled individually after a second multidisciplinary discussion.“
“5. Is there place for ChRT without adding CHT in neoadjuvant treatment protocols( if you have good response, MRF- after ChRT).”
Answer: Our current algorithm does not include chemoradiation alone as neoadjuvant treatment option anymore for patients capable of receiving TNT. Of course, this might be an option after individual multidisciplinary discussion for patients with severe comorbidity which are deemed not suitable for a doublet chemotherapy regimen. However, for patients in good shape, the major advantage of shifting the chemotherapy into the neoadjuvant phase is not only a locoregional or “surgical” benefit driven by improved response. The major benefit seems to be the reduced distant metastases rate and consequently the improved disease-free survival, which is probably based on a better eradication of micrometastases either by earlier application of chemotherapy and/or improved dose-intensity by improved treatment compliance compared to the adjuvant approach. Therefore, we feel that all high and most intermediate risk patients should receive neoadjuvant chemotherapy regardless of their response to upfront chemoradiation or short course RT, because most of them will otherwise need adjuvant chemotherapy which is far less tolerated.
“6. Is there place for neoadjuvant CHT without RT in neoadjuvant treatment? Most clear indication for neoadjuvant treatment for rectal cancer is necessity of tumor downstaging and downsizing ( MRF+ or sphincter sparing for low cancers) and this has to be discussed individually with patient and in MDT meeting. So in my personal opinion there is possibility of overtreatment with this TNT approach for many groups of patients, according this concept. But that's just my personal opinion as mentioned above, the article is really useful and valuable.”
Answer: We thank the reviewer for sharing his/her view, although we do have a differing opinion in this regard: first, we are unsure if there is sufficient data to justify an upfront chemotherapy only approach in locally advanced-rectal cancer patients. Most patients with enhanced risk for distant failure share also factors with increased risk for locoregional failure. Moreover, no chemotherapy regimen alone has been tested against TNT approaches, which represent the best available evidence regarding outcome in locally-advanced rectal cancer patients. Similar to our answer regarding the role of chemoradiation alone as neoadjuvant treatment, we feel that the most important benefit from TNT approaches is not locoregional response but reduction of distant failure and improved DFS.
We agree with the reviewer that TNT includes some risk of overtreatment because assessing the risk profile solely on imaging will never be completely accurate. However, in the light of a possible disease- free survival advantage (as shown by two randomized trials), the risk of undertreating patients seems even more difficult to explain, especially if the burden (and limited efficacy) of salvage treatments (like second line systemic therapy, surgery or RT) in case of distant failures are taken into account.

Reviewer 2 Report
Dear Authors,
Thank you for your submission.
Comments:
The Authors provided an evidence-based institutional opinion and proposed a paradigm for neoadjuvant therapy in localized rectal cancer.
Therefore, this issue would be addressed in the title such as "-An institutional opinion for neoadjuvant therapy " or "-An institutional paradigm for neoadjuvant therapy" or “"-A proposed paradigm for neoadjuvant therapy”
The authors have recommended a fixed dose of 50.4 Gy for T3 and T4 tumors. Why did they not consider an escalated dose up to 54 Gy for T4 tumors?
While the authors focus on “wait and watch” as the preferred option for distal rectal tumors, there is a paucity of recommendations for radiation dose intensification in these tumors to achieve a higher rate of CPR.
As well as, the authors have recommended a fixed 6 cycles of CAPEOX chemotherapy regimen for all patients. The intensity and duration of chemotherapy can be flexible based on the disease stage and the patient's performance status. This issue should be explained in the figures and discussion section.
The treatment option for CT1-2CN0 distal tumors is vague. The authors have recommended surgery (local excision?) followed by adjuvant CHT according to pTNM. According to the current guidelines such as NCCN, high-risk T1 and T2 lesions should be treated by APR or local excision followed by adjuvant CRT. Adjuvant chemotherapy is usually considered for patients with residual disease after CRT.
Author Response
The Authors provided an evidence-based institutional opinion and proposed a paradigm for neoadjuvant therapy in localized rectal cancer. Therefore, this issue would be addressed in the title such as "-An institutional opinion for neoadjuvant therapy " or "-An institutional paradigm for neoadjuvant therapy" or “"-A proposed paradigm for neoadjuvant therapy”
Answer: We agree with the reviewer and have changed the title to “How we treat localized rectal cancer - An institutional paradigm for total neoadjuvant therapy”
“The authors have recommended a fixed dose of 50.4 Gy for T3 and T4 tumors. Why did they not consider an escalated dose up to 54 Gy for T4 tumors? While the authors focus on “wait and watch” as the preferred option for distal rectal tumors, there is a paucity of recommendations for radiation dose intensification in these tumors to achieve a higher rate of CPR. As well as, the authors have recommended a fixed 6 cycles of CAPEOX chemotherapy regimen for all patients. The intensity and duration of chemotherapy can be flexible based on the disease stage and the patient's performance status. This issue should be explained in the figures and discussion section.”
Answer: We generally agree with the reviewer that even more personalization of the treatment would have been possible, for example with regard to total radiation dose, number of chemo cycles and so on. However, we feel that we have already included a fair amount of individualization into our concept and at least some factors should be kept stable. We therefore decided to use the chemoradiation and chemotherapy strategies with the largest existing evidence regarding efficacy and toxicity. To our opinion, this is chemoradiation with a total dose of 50.4 Gy (which has been used in mainly all recent trials prior to TNT approaches) and for chemotherapy with 6 cycles of CAPOX (which has been used in the RAPIDO trial, which is the largest TNT trial). We feel that the combination of chemoradiation with 50.4 Gy and 6 cycles of CAPOX is already a considerable (and sufficient) treatment intensification compared to the prior standard of chemoradiation alone.
We also agree with the reviewer that further radiation dose escalation will likely impact response based on the general radiobiological models and there is some clinical data indicating higher cCR rates with radiation dose intensification by EBRT, brachytherapy or contact x-ray therapy (as similarly suggested for intensification of simultaneous chemotherapy during radiation by adding oxaliplatin). However, most of these studies were small, did not focus on certain risk factors, and did not use their intensified chemoradiation concept within a TNT approach. Moreover, most of these studies were not able to show a clear and distinct benefit in outcome within their range of dose intensification (mainly less than 10% in total dose) compared to standard radiation dose in the absence of randomization or comparative arms. In contrast, studies using more intensive regimens (for example Appelt et al. using 60 Gy + brachytherapy) reported increased toxicities. We therefore decided during our discussions to further use our well established chemoradiation concept with 50.4 Gy and capecitabine. It is similar to the regimen used in the OPRA predecessor study (Garcia-Aguilar et al.) and was also used within the OPRA trial (50,4 to 56 Gy allowed, median dose 54 Gy observed in both arms). During our discussions, we evaluated using the observed median dose from the OPRA trial (54 Gy) for distal tumors or for all high risk tumors according to our definition. However, we decided that such a small dose escalation (roughly 7%) for patients with non-distal tumors (who are scheduled for surgery anyway) does not seem to be meaningful. In distal tumors we weighed the possible small increase in cCR (if any) versus the difficulties in having another different scheme within our clinical routine and decided to keep the total radiation dose constant at 50.4 Gy for all patients with chemoradiation.
We also agree with the reviewer that further intensification of chemotherapy might be beneficial for some patients with regard to cCR. However, we decided not to use the FOLFIRINOX scheme from the PRODIGE23 trial because we deemed it too intensive for most of our patients without a clear benefit by cross trial comparison. We instead decided to use the well-established concept of 6 cycles CAPOX from the RAPIDO trial for all patients receiving neoadjuvant chemotherapy (which is also the maximum cycles of CAPOX given within the OPRA trial). We are not aware of data supporting better cCR rates with more cycles of CAPOX, therefore we do not support further intensification. Regarding de-intensification based on stage, we do not feel that there is adequate data supporting a shorter concept without risking the benefits (especially in distant failure reduction) shown in the RAPIDO trial. Of course, we take performance status into account prior to scheduling patients for different treatments, however the described algorithm was not designed for frail or severely comorbid patients, which in our opinion usually do not qualify at all for TNT approaches. Patients are treated according to this algorithm in our institution if they are felt to be able to tolerate the recommended treatment, otherwise they are discussed in our multidisciplinary board and are treated in a highly individualized fashion. If patients develop toxicity or loss of performance during therapy, treatment is of course adapted if needed, however this seems a matter of clinical routine and may not be necessarily described within a general treatment algorithm.
“The treatment option for CT1-2CN0 distal tumors is vague. The authors have recommended surgery (local excision?) followed by adjuvant CHT according to pTNM. According to the current guidelines such as NCCN, high-risk T1 and T2 lesions should be treated by APR or local excision followed by adjuvant CRT. Adjuvant chemotherapy is usually considered for patients with residual disease after CRT.”
Answer: We agree with the reviewer that we have not included our detailed policy for cT1/T2 cN0 tumors. This is because the treatment of early-stage tumors is mainly beyond the scope of our article (which focuses on neoadjuvant treatment of rectal cancer as indicated in the title). We have included those stages into our algorithm of distal tumors only to show that we will offer TNT also to early stage patients if sphincter sparing surgery is not possible. Because it would be difficult to include also a detailed substantiation of our treatment principles for early stage tumors of any location into the main text (local excision versus transabdominal resection, adjuvant treatment based on pTNM and/or risk features), we would prefer not to include it substantially into the figures. However, for clarification purposes, we have changed figure 2 for upper, all other: surgery > adjuvant C(R)T based on final pathology), mid (cT1-3b cN0: surgery > adjuvant C(R)T based on final pathology) and distal tumors (cT1-2 cN0 with sphincter sparing surgery possible: surgery > adjuvant C(R)T based on final pathology). We have also included a footnote into figure 2 and 3, that in case of T1N0 tumors local excision is possible if appropriate.

Reviewer 3 Report
The present study focuses on a new treatment regime for localized rectal cancer. The authors do a superb job in presenting their protocol, considering the benefits and pitfalls of already established treatment regimes for localized rectal cancer. The algorithm encompasses microsatellite instability that is not normally explained in as such. Overall, I commend the authors on a well-structured article.
Author Response
We thank the reviewer for his/her comments.

Round 2
Reviewer 2 Report
Dear Authors,
Thank you so much for your revision.